# Photocatalytic radical defluoroalkylation of unactivated alkenes via distal heteroaryl *ipso*-migration

Xin Yuan[1,2], Kai-Qiang Zhuang[1,2], Yu-Sheng Cui[1,2], Long-Zhou Qin[1,2], Qi Sun[1,2], Xiu Duan[1,2], Lin Chen[1,2], Ning Zhu[1,2], Guigen Li[3,4], Jiang-Kai Qiu [1,2✉] & Kai Guo [1,2✉]

Currently, the selective activation of $C(sp^3)$–F bonds and C–C bonds constitute one of the most widely used procedures for the synthesis of high-value products that range from pharmaceuticals to agrochemical applications. While numerous examples of these two methods have been reported in their respective fields, the processes which merge the activation of both single $C(sp^3)$-F bonds and C–C bonds in one step still remain elusive. Here, we demonstrate the controllable defluoroalkylation–distal functionalization of trifluoromethylarenes with unactivated alkenes via distal heteroaryl migration. This is proposed to proceed via tandem $C(sp^3)$–F and C–C bond cleavage using visible-light photoredox catalysis combined with Lewis acid activation. This strategy provides facile and flexible access to multiply functionalized $\alpha,\alpha$-difluorobenzylic ketones in useful yields (up to 88%) under mild conditions. The products can be further transformed into other valuable compounds, demonstrating the method's utility.

[1] Biotechnology and Pharmaceutical Engineering, Nanjing Tech University, Nanjing 211816, P. R. China. [2] State Key Laboratory of Materials-Oriented Chemical Engineering, Nanjing Tech University, Nanjing 211800, P. R. China. [3] Institute of Chemistry & Biomedical Science, Nanjing University, Nanjing 210093, P. R. China. [4] Department of Chemistry and Biochemistry, Texas Tech University, Lubbock, TX 79409-1061, USA. ✉email: qiujiangkai@njtech.edu.cn; guok@njtech.edu.cn

To address environmental concerns and achieve high step economy, the activation of C(sp³)–F bonds is one of the most green and efficient methods for accessing target fluorine compounds[1]. Therefore, the development of novel and efficient synthetic approaches for the direct functionalization of unactivated C(sp³)–F bonds from easily available reagents such as trifluoromethylarenes (ArCF₃) is of vital importance. However, the reactivity and selectivity of this process are limited due to the high energy of C(sp³)–F cleavage (~115 kcal/mol for PhCF₃) and the shielding effect of the three F atoms[2]. Conventional methods for the cleavage of the C–F bonds in ArCF₃ include electrochemical reduction[3,4], the use of low-valent metals[5–7], and the application of frustrated Lewis pairs[8,9]. However, due to the gradual decrease in the strength of the remaining C(sp³)–F bonds (99 kcal/mol for PhCFH₂), it becomes exceedingly difficult to avoid multiple defluorinations[10]. Compared to examples in which all three C(sp³)–F bonds in ArCF₃ are cleaved without selectivity, few examples of single C(sp³)–F bond cleavage in ArCF₃ have been reported[11]. Importantly, several appealing strategies have been established for the selective cleavage of a single C(sp³)–F bond in Ar–CF₃ substrates, enabling efficient access to valuable

ArCF₂R derivatives (Fig. 1a). For instance, Yoshida and co-workers demonstrated the cleavage of a single C(sp³)–F bond accompanied by the transformation of F atom into an ortho-silylium cation, providing an aryldifluoromethyl cation that can react with various nucleophilic species[12]. Recently, Bandar's group reported a fluoride-initiated sequential allylation/derivatization reaction for the construction of diverse α,α-difluorobenzylic compounds[13]. Photocatalysis has also found applications in this field. Gschwind and König disclosed the photocatalytic single C(sp³)–F functionalization of ArCF₃ with N-aryl acrylamides via photocatalysis combined with Lewis acid activation[14]. Moreover, Jui and co-workers reported the photoredox-catalyzed intermolecular defluorinative coupling of ArCF₃ with unactivated alkenes[15,16]. Despite these achievements, general and mild strategies for defluoroalkylation that allow the rapid activation of single C(sp³)–F bonds from easily available ArCF₃ remain elusive.

On the other hand, the cleavage of inert C–C single bonds has enriched the synthetic arsenal for the synthesis of complex bioactive molecules[17]. In contrast to conventional transition-metal catalysis, which is restricted by harsh reaction conditions

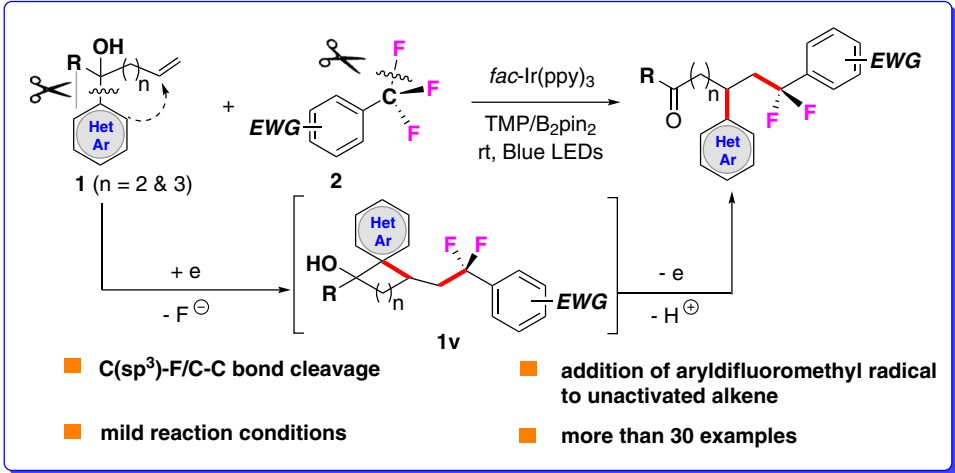

**Fig. 1 Selective functionalization of C(sp³)–F and C–C bonds. a** Monoselective C(sp³)–F functionalization. **b** Radical triggered C–C bond cleavage.
**c** Selective dual C(sp3)–F/C–C bond cleavage.

**Table 1 Optimization of the reaction conditions[a].**

| Entry | Deviation from the standard conditions | Yield (%)[b] |
|---|---|---|
| 1 | None | 84 |
| 2 | **PC2–PC8** as the catalyst instead of *fac*-Ir(ppy)₃ | N.D. |
| 3 | Quinuclidine as the amine | 17 |
| 4 | HBpin as the F—Scavengers | 42 |
| 5 | THF as the solvent instead of DCE | 66 |
| 6 | DMPU as the solvent instead of DCE | 50 |
| 7 | MeOH as the solvent instead of DCE | 48 |
| 8 | 0.2 mL H₂O as an additive | 30 |
| 9 | Without *fac*-Ir(ppy)₃ | N.R. |
| 10 | Without light | N.R. |

[a]Standard conditions: **1a** (2.0 eq., 0.2 mmol), **2a** (1.0 eq., 0.1 mmol), *fac*-Ir (ppy)₃ (1.0 mol%), TMP (3.0 eq., 0.3 mmol), B₂Pin₂ (3.0 eq., 0.3 mmol), DCE 1.0 mL, 25 °C, N₂, 455 nm, 24 h.
[b]Isolated yield is based on **2a**. *N.R.* no reaction, *N.D.* not detected.

and the need for costly catalysts, controllable radical-medicated C–C bond activation overcomes the above problems of transition-metal catalysis while providing excellent atom and step economies[18]. In particular, the radical-triggered cleavage of C–C bonds via heteroaryl group migration is an attractive approach in modern organic synthesis[19–24]. Interestingly, Zhu and co-workers recently reported the radical-triggered fragmentation of unstrained C–C bonds via the migration of distal functional groups (Fig. 1b)[25,26]. Meanwhile, we reported the sulfonyl radical-triggered difunctionalization of alkenes via remote heteroaryl *ipso*-migration under electrochemical conditions[27]. Prompted by these results, our group aims to develop novel and general methods to synthesize valuable aryldifluoromethyl derivatives (ArCF₂R) via selective C(sp³)–F and C–C bond activation under mild conditions. Since ArCF₂R compounds can be obtained from available ArCF₃ compounds via selective single C(sp³)–F bond cleavage, we questioned whether the aryldifluoromethyl radical (ArCF₂) generated photocatalytically in combination with Lewis acid activation could be captured by the unactivated olefins on the tertiary alcohol **1** and trigger the distal migration of the heteroaryl group via the cyclic intermediate **1v** followed by oxidation and deprotonation to give the corresponding α,α-difluorobenzylic ketones (Fig. 1c). Nevertheless, to the best of our knowledge, the sequential activation of an inert C(sp³)–F/C–C bond-triggered multiple-cascade process for the divergent construction of complex skeletons has not been reported. Herein, we report the development of such a transformation that involves tandem C(sp³)–F and C–C bond functionalization via visible-light photoredox catalysis together with Lewis acid activation.

## Results

**Reaction optimization**. We commenced our investigation by choosing 1-(benzo[*d*]thiazol-2-yl)-1-phenylpent-4-en-1-ol (**1a**) and 4-trifluoromethylbenzonitrile (**2a**) as model substrates to test the reaction conditions. To our delight, by using *fac*-Ir(ppy)₃ [for Ir(III)/Ir(II), $E_{red} = -2.19$ V vs. saturated calomel electrode (SCE)][14] as the photocatalyst, bis(pinacolato)diboron (B₂pin₂) and 2,2,6,6-tetramethylpiperidine (TMP) as additives, and 1,2-dichloroethane (DCE) as the solvent, the desired heteroaryl-migrated product **3a** was obtained in 84% yield (Table 1, entry 1). X-ray diffraction analysis of the heteroaryl-migrated product of **3a** confirmed the structural assignment of the reaction products (for

details, see the Supplementary Fig. 2). Other transition-metal photocatalysts (**PC1–PC8**), including Ir(ppy)₂(dtbbpy)PF₆ (**PC1**), Ir[{dF(CF₃)ppy}₂(dtbbpy)]PF₆ (**PC2**), and Ir(dmppy)₂(dtbbpy)PF₆ (**PC3**), were not suitable for this transformation (Table 1, entry 2). It should be noted that the combination of B₂pin₂ and TMP was indispensable for this reaction, a lower reaction efficiency was observed in the absence of either B₂pin₂ or TMP (Table 1, entries 3 and 4). The effect of the solvent was also explored. Compared to DCE, the use of THF (Tetrahydrofuran), DMPU (1,3-Dimethyl-Tetrahydropyrimidin-2(1*H*)-one), and MeOH as solvents resulted in lower yields (Table 1, entries 5–7). However, the addition of 0.2 mL of water led to a decrease in reaction yield (Table 1, entry 8). Furthermore, the Ir catalyst and blue-light irradiation were crucial to this transformation, as evidenced by the dramatic decrease in efficiency when the reaction was carried out without either the Ir catalyst or blue-light irradiation (Table 1, entries 9 and 10). Finally, under the optimum conditions, the reaction of **1a** (2.0 equiv), **2a** (1.0 equiv), *fac*-Ir(ppy)₃ (1.0 mol%), B₂pin₂ (3.0 equiv), and TMP (3.0 equiv) in DCE at 25 °C under irradiation from a 50-W blue-light-emitting diode (455 nm) for 24 h provided the desired product **3a** in 84% isolated yield.

**Evaluation of substrate scope**. With the optimized reaction conditions in hand, we expanded the scope of this defluoroalkylation–distal functionalization strategy. First, different substituted benzothiazole tertiary alcohols **1** were investigated as substrates to react with 4-trifluoromethylbenzonitrile (**2a**) (Fig. 2). Notably, the electronic character of the aryl group (R¹) did not have a strong effect on the reaction outcome, a range of substituents including electron-neutral (Me, *t*Bu), electron-donating (MeO), and electron-withdrawing (F, Br, and CF₃) substituents on the *para*-positions of the aromatic rings were well tolerated and resulted in the corresponding products with yields ranging from 30 to 66% (**3b–3g**). Meanwhile, substituents on the *ortho*- and *meta*-positions of the aromatic rings were also compatible under the optimized reaction conditions (54% yield for **3h**, 35% yield for **3i**, and 44% yield for **3j**). In addition to phenyl derivatives, other aryl groups such as thienyl (**1k**) and naphthyl (**1l**) groups were suitable for this intramolecular heteroaryl migration, although lower yields were obtained (**3k** and **3l**). Subsequently, we turned our attention to the generality of the migrating groups. A variety of *N*-containing heteroaryl groups

**Fig. 2 Scope of heteroaryl-substituted tertiary alcohols as substrates.** Reaction conditions: **1** (2.0 eq., 0.2 mmol), **2a** (1.0 eq., 0.1 mmol), *fac*-Ir(ppy)$_3$ (1.0 mol%), TMP (3.0 eq., 0.3 mmol), B$_2$Pin$_2$ (3.0 eq., 0.3 mmol), THF 1.0 mL, 25°C, N$_2$, 455 nm, 24 h. Isolated yield is based on **2a**.

were tested for their migratory aptitude (**1m–1r**). The *N*-containing heteroaryl groups could also be extended to benzoxazole (**1o**), pyridine (**1p**), and thiazole (**1q** and **1r**), although with slightly decreased yields (**3o–3r**).

To further expand the scope of this transformation, we examined the other reaction partner, electron-deficient Ar–CF$_3$ systems, using benzothiazole-substituted tertiary alcohol **1a** as the trapping reagent. As shown in Fig. 3, the *para*-cyano-substituted benzotrifluorides were good substrates for this reaction, regardless of some electron-withdrawing (F and Cl) substitutions located on the *ortho*- or *meta*-positions of the aromatic rings (77% yield for **4a**, 82% yield for **4b**, and 84% yield for **4c**). In contrast to the *para*-cyano-substituted benzotrifluorides, the *ortho*- and *meta*-cyano-substituted benzotrifluorides were not compatible with this migration process, and low chemical efficiency was observed (**4e** and **4f**). The trifluoromethylaromatics bear some other electron-withdrawing groups in addition to the cyano group, including sulfonyl and ester groups. 4-Trifluoromethylbenzenesulfonyl protected acyclic secondary amines, synthetically useful heterocycles (morpholine, glycine

derivative), also proved to be appropriate substrates, affording the migrated products in moderate to good yields (**4g–4k**). Methyl 4-(trifluoromethyl)benzoate (**1n**) exhibited low reactivity, giving the desired product in less than 15% yield (**4n**). Pyridine substrates with trifluoromethyl groups at the 2- or 3-positions did not give rise to the corresponding pyridine-migrated products (**4o–4q**).

**Proposed mechanism.** Several control experiments were performed to gain insight into the mechanistic details of this defluoroalkylation–distal functionalization system (Fig. 4). First, the radical trapping agent 2,2,6,6-tetramethylpiperidine-1-oxyl (TEMPO) or butylated hydroxytoluene (BHT) was added under the standard reaction conditions with benzothiazole-substituted alkene **1a** and 4-trifluoromethylbenzonitrile (**2a**) (Fig. 4a, b). As expected, the desired product **3a** was obtained only in a trace amount accompanied by the corresponding trapping adducts **5a** and **6**. This indicates that the aryldifluoromethyl radical is involved in this transformation (for details, see the Supplementary Figs. 3–5). Using diphenylethylene **7** as trapping reagent, the CF$_2$-adduct **8** was detected by HRMS (Fig. 4c). Next,

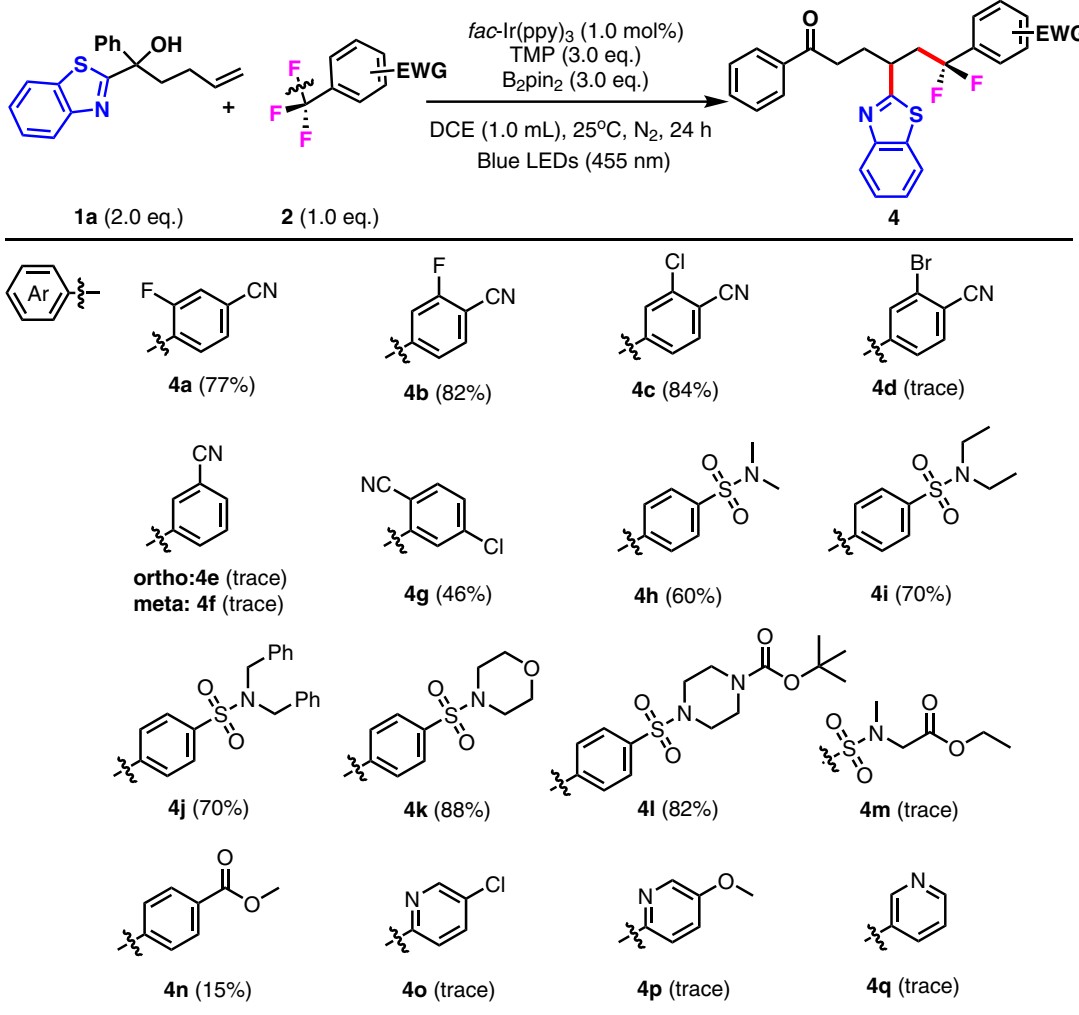

**Fig. 3 Substrate scope for Ar-CF₃ systems.** Reaction conditions: **1a** (2.0 eq., 0.2 mmol), **2** (1.0 eq., 0.1 mmol), *fac*-Ir(ppy)₃ (1.0 mol%), TMP (3.0 eq., 0.3 mmol), B₂Pin₂ (3.0 eq., 0.3 mmol), DCE 1.0 mL, 25 °C, N₂, 455 nm, 24 h. Isolated yield is based on **2**.

TMP (0.3 mmol) and B₂Pin₂ (0.3 mmol) were mixed and dissolved in CD₂Cl₂. A new boron species ($^{11}$B NMR, $\delta = 24.26$ ppm in CD₂Cl₂) was observed at a relatively low concentration (for details, see the Supplementary Fig. 8). Based on previous reports[21], we assumed that the barium cation **K** was generated in situ from the reaction of TMP with B₂Pin₂, which might abstract an F anion from the radical anion **B**. Based on the Stern–Volmer relationship, we determined that 4-trifluoromethylbenzonitrile quenched excited *fac*-*Ir(ppy)₃ (for details, see the Supplementary Figs. 11 and 12). We also demonstrated the further elaboration of an aryl migrated product (Fig. 5).

To understand the effect of the length of the tethered alkyl chain on the *ipso*-heteroaryl migration, a range of benzothiazole-substituted tertiary alcohols **1** with different chain lengths were applied in the reaction under the standard conditions (Table 2). Among the tested tertiary alcohols, only bishomoallylic alcohol (**1a**, $n = 2$) and trishomoallylic alcohol (**1t**, $n = 3$) afforded the corresponding heteroaryl-migrated products (**3a** and **3t**, respectively), indicating that this radical-induced heteroaryl migration process might involve cyclic transition states. The migration process prefers the thermo-dynamically favored five- and six-membered cyclic transition states ($n = 2$ and 3) over the four- and seven-membered cyclic transition states ($n = 1$ and 4).

Based on the above reuslts suggesting the trapping of intermediates and related reports in the literature, we propose the following tentative mechanism for the reaction (Fig. 6). Under visible light, Ir(ppy)₃ is excited to Ir(ppy)₃* [$E_{1/2}$ IV/III* = −1.73 V vs. SCE in MeCN][28]. Subsequently, a reduced iridium complex (Ir²⁺) is generated along with the the TMP radical cation **A** via quenching reduction. The reduced Ir²⁺ acts as a reductant to reduce 4-trifluoromethylbenzonitrile (**2a**) to the corresponding aryldifluoromethyl radical **D**, and Ir³⁺ is regenerated via SET. At the same time, the Lewis-acidic barium cation **K** is produced by the reaction between the protonated TMP species **C** and B₂Pin₂, which abstracts F⁻ from **B** to give the aryldifluoromethyl radical **D**. The benzothiazole-substituted tertiary alcohol **1** captures **D** to form the alkyl radical intermediate **E**, which is intercepted by the C=N double bond of the heteroaryl group via a five-membered cyclic transition state to give the spiro-bicyclic N-centered radical intermediate **F**. The amino radical triggers C–C bond cleavage, and the resultant ring opening of the spiro structure generates the thermodynamically favored ketyl radical **G**. The single-electron oxidation of **G** to the cationic intermediate **J** and the subsequent deprotonation afford the final product **3a**.

In summary, we have demonstrated a novel and efficient method for the selective cleavage of C(sp³)–F and C–C bonds toward the defluoroalkylation of trifluoromethylaromatic substrates with unactivated alkenes via distal heteroaryl migration.

**a) Trapping experiment by TEMPO**

**1a** (2.0 eq.)          **2a** (1.0 eq.)

standard conditions

TEMPO (3.0 eq.)

**3a** (N.D.)

**b) Trapping Experiment by BHT**

**1a** (2.0 eq.)          **2a** (1.0 eq.)

standard conditions

BHT (3.0 eq.)

**3a** (N.D.)

**c) Trapping Experiment by diphenylethylene**

**2a** (1.0 eq.)          **7** (3.0 eq.)

standard conditions

**8** (saturated) + **8'** (unsaturated)
8% ($^{19}$F NMR yield)

**Fig. 4 Mechanistic studies.** Standard conditions: *fac*-Ir(ppy)$_3$ (1.0 mol%), TMP (3.0 eq., 0.3 mmol), B$_2$Pin$_2$ (3.0 eq., 0.3 mmol), THF 1.0 mL, 25 °C, N$_2$, 455 nm, 24 h. **a** Trapping experiment by TEMPO. **b** Trapping experiment by BHT. **c** Trapping experiment by diphenylethylene.

1. Me$_3$OBF$_4$, DCM
2. NaBH$_4$, MeOH
3. AgNO$_3$, MeCN/H$_2$O

**3a**

**9 (32% over 3 steps)**

**Fig. 5 Further elaboration of 3a.** The corresponding aldehyde **9** can be obtained over 3 steps (32%, the isolated yield is based on **3a**).

| Table 2 Effect of chain length on reaction efficiency[a]. |
|---|

**1** (2.0 eq.)          **2a** (1.0 eq.)

*fac*-Ir(ppy)$_3$ (1.0 mol%)
TMP (3.0 eq.)
B$_2$pin$_2$ (3.0 eq.)

DCE (1.0 mL), 25°C, N$_2$
455 nm, 24 h

**3**

| Compounds | 3r (n = 0) | 3s (n = 1) | 3a (n = 2) | 3t (n = 3) | 3u (n = 4) |
|---|---|---|---|---|---|
| TS ring size | 3 | 4 | 5 | 6 | 7 |
| Yield (%) | 0 | 0 | 84 | 59 | 0 |

[a]Cited yields are for the isolated material following chromatography. Reaction conditions: **1** (2.0 eq., 0.2 mmol), **2a** (1.0 eq., 0.1 mmol), *fac*-Ir(ppy)$_3$ (1.0 mol%), TMP (3.0 eq., 0.3 mmol), B$_2$Pin$_2$ (3.0 eq., 0.3 mmol), DCE 1.0 mL, 25 °C, N$_2$, 455 nm, 24 h.

The inert C(sp$^3$)–F and C–C bonds are readily cleaved in sequence based on the combination of visible-light photocatalysis and Lewis acid activation. A range of α,α-difluorobenzylic ketones was obtained with complete selectivity in moderate to good yields.

The reaction features mild conditions and broad functional-group tolerance. Further studies on radical cascade reactions for the construction of difluoromethylated derivatives and other high-value product classes are currently ongoing in our laboratory.

**Fig. 6 Plausible mechanism. a** A proposed photocatalytic cycle. **b** The production of Lewis-acidic barium cation **K**. **c** The C–C bond cleavage and the ring opening of the spiro structure.

## Methods

**General information**. For more details, see Supplementary Fig. 1 and the Supplementary Methods.

**X-ray crystallography structure of compounds 3a**. For the CIF see Supplementary Data 1. For more details, see Supplementary Fig. 2 and Table 1.

**Detailed optimization of the reaction conditions**. For more details, see Supplementary Tables 2–5.

**Mechanistic investigation**. For more details, see Supplementary Figs. 3–8.

**Fluorescence quenching experiment**. For more details, see Supplementary Figs. 9–16.

**Synthesis and characterization**. See Supplementary Methods (general information about chemicals and analytical methods, synthetic procedures, product derivation, $^1$H and $^{13}$C NMR data, and HRMS data), Supplementary Figs. 17–105 ($^1$H and $^{13}$C NMR spctra).

**General procedure for the synthesis of 3a**. 1-(benzo[d]thiazol-2-yl)-1-phenyl-pent-4-en-1-ol **1a** (0.2 mmol, 59.0 mg, 2.0 eq.), 4-(Trifluoromethyl)benzonitrile **2a** (0.1 mmol, 17.1 mg, 1.0 eq.), and *fac*-Ir(ppy)$_3$ (0.001 mmol, 0.7 mg, 1.0 mol%) were added into a 25 mL snap vial equipped with a stirring bar. The vial was purged with N$_2$ for three times via syringe needle. Then TMP (0.3 mmol, 42.3 mg, 51 μL, 3.0 mg), dry THF (1.0 mL) and B$_2$pin$_2$ (0.3 mmol, 76.2 mg, 3.0 eq.) were added sequentially by syringes. Then the reaction mixture was irradiated through the

bottom side of the vial by Blue LEDs at 25 °C. All the reaction was stopped, the mixture was transferred into a separating funnel and diluted by DCM (30 mL). The organic layer was washed by H$_2$O (10 mL × 2) and brine (10 mL), dried over anhydrous Na$_2$SO$_4$, and then concentrated under reduced pressure. The resulting residue was purified by flashed column chromatography to obtain the desired product.

## Data availability

The X-ray crystallographic coordinates for structures reported in this article have been deposited at the Cambridge Crystallographic Data Centre (CCDC), under deposition number CCDC 1867225 (**3a**). These data can be obtained free of charge from The Cambridge Crystallographic Data Centre via www.ccdc.cam.ac.uk/data_request/cif. The data supporting the findings of this study are available within the paper and its Supplementary Information (Supplementary Data 1—crystallographic information file for compound **3a**). All relevant data are also available from the authors.

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

## Acknowledgements

We are grateful for financial support from the National Natural Science Foundation of China (Grant nos. 21702103, 21522604, U1463201, and 21402240); the youth in Jiangsu Province Natural Science Fund (Grant nos. BK20150031, BK20130913, and BY2014005-03).

## Author contributions

X.Y. performed the experiments and analyzed the data. K.Z. perpared the substrate scope **1f** and **1g**. Y.C. perpared the substrate scope **1a–1e**. L.Q. modified the supporting information. Q.S. and X.D. modified the paper. L.C. and N.Z. checked the format of the paper. G.L. and K.G. provided the money and laboratory platform. J.Q. designed the project and wrote the original draft. All of the authors discussed the results and commented on the paper.

## Competing interests

The authors declare no competing interests.
