## [Peer Review file · Communications Chemistry]

Reviewers' comments:

Reviewer #1 (Remarks to the Author):

Guo and coworkers report a strategy for merging photoredox-mediated defluorofunctionalization of trifluoromethylarenes with heteroaryl migration to produce difluorobenzyl ketone products. The reaction employs conditions similar to König's work on the addition of trifluoromethylarenes to N-aryl acrylamide acceptors, but instead uses alkene acceptors that transfer N-heteroaryl groups to the radical adduct. The reaction proceeds under mild conditions but requires 2 equivalents of the alkene acceptor. Mechanistically, this is a nice extension of previous work in photoredox-enabled trifluoromethylarene defluoroalkylation chemistry and should expand the scope of accessible difluorobenzyl products. The authors show a range of heteroaryl groups that transfer (primarily 1,3-azole derivatives) and also demonstrate a few types of trifluoromethylarenes that this reaction works for. Mechanistic studies are included that support the proposed SET and radical-based mechanism.

Overall, this work is a welcomed extension of trifluoromethylarene functionalization chemistry but the extremely limited scope of ArCF₃ substrates reduces the utility of this method in its current state. Only 4-cyano and 4-sulfonamide benzotrifluorides participate in this reaction; this is a much narrower scope than reported by König or Jui, and it is unclear why this reaction does not work for other substrate classes. The authors should either expand the scope of ArCF₃ partners or show that their products can be further derivatized. In the abstract, they state that the "products could be further transformed into other valuable compounds" but do not show this at all. I believe one of these two improvements should be made before publishing this work.

Some additional comments and questions from the manuscript and supporting information:

For the radical trapping experiments shown in Figure 4, it would be very helpful to include what products were observed and in what quantity. Only stating that the coupling product was not obtained is of limited value, and including approximate yields of trapped intermediates would also be useful (compared to just stating that the adduct was "detected by HRMS"). Similarly, is F-Bpin observed as a byproduct of this reaction?

The authors state on line 205 that the reaction features "unique regioselectivity/stereoselectivity". What does this mean? I don't think this reaction has any stereochemical features to it at all and "unique regioselectivity" is meaningless if it is not referenced to a specific feature/other methods.

In the Supporting Information, it would be helpful to include the splitting patterns observed in the F NMR spectra and to include the IR data of the products.

Reviewer #2 (Remarks to the Author):

In this manuscript, Guo and co-workers have disclosed a photocatalytic radical defluoroalkylation of electron-deficient trifluoromethylarenes of unactivated alkenes via distal heteroaryl ipso-migration, which is originally developed by Zhu (JACS, 2017, 139, 1388). Surprisingly, this seminal paper was not cited herein. Frankly speaking, this work is a smart combination of two interesting concepts, C(sp³)-F activation and remote functional group migration. As a result, the reaction achieves a sequence of C-F and C-C bonds scission under mild conditions, which are generally challenging issues in synthetic chemistry. The substrate scope is reasonably broad, and some experiments have been performed to elucidate the possible mechanism. Because the overall transformation merges two current hot topics, it would attract broad interest from synthetic chemists. In sum, this manuscript could be considered for publication in Commun. Chem. but after major revisions.

1. Page 2, Line 44, ref. 14. Gschwind and König's work should be corrected with JACS 2017, 139, 18444.

Page 2, Line 46, ref. 15. The related literature on photoredox-catalyzed monoselective defluorinative functionalization should also be corrected: JACS 2019, 141, 13203.

Page 4, Line 57-61, the seminal literatures or reviews on distal heteroaryl migration should be cited, e.g. JACS 2017, 139, 1388; ACIE 2018, 57, 1640; ACIE 2018, 57, 17156; Org. Chem. Front. 2018, 5, 1896; Tetrahedron Lett. 2018, 59, 1328; CJC, 2019, 37, 171, etc.

2. There are many grammatical errors and typos. The manuscript should be thoroughly polished. For example:

Page 7, Line 116, "thiophene (1k) or naphthalene (1l)" should be changed to "thienyl (1k) or naphthyl (1l)", and related description should also be corrected in Fig. 2.

Page 8, line 122, the compound is "benzoxazole (1o)" not "benzimidazole (1o)".

Other format errors in Page 17, Line 258 and Line 265; Page 18, Line 281 should be corrected.

3. Page 7, Line 120, no electron-withdrawing substituents have been tested, so the conclusion "the substituted benzothiazoles were migrated smoothly in spite of the electronic characters" is not accurate.

4. Fig 5, the authors should explain the step from G to J. Which species acts as the single-electron oxidant?

Reviewer #3 (Remarks to the Author):

In the manuscript, the authors reported a tandem C(sp³)-F and C-C bonds functionalization by visible-light photoredox catalysis and Lewis acid activation. The strategy provides a useful method for construction of multiple functionalized α , α -difluorobenzyl ketone in moderate to good yields under mild conditions. The authors also proposed a reasonable mechanism for this transformation. This is a meaningful research.

However, The English writing should be polished, and some spelling and grammatical mistakes need to be corrected. For example, 'theombination' in Line 202 might be 'the combination'.

In short, this manuscript is required an appropriate revision before to be published.

Dear reviewers:

Thank you very much for your valuable suggestion on our manuscript entitled "Photocatalytic radical defluoroalkylation of unactivated alkenes *via* distal heteroaryl *ipso*-migration" (COMMSCHEM-19-0372A) by Xin Yuan, Kai-Qiang Zhuang, Yu-Sheng Cui, Long-Zhou Qin, Qi Sun, Xiu Duan, Lin Chen, Ning Zhu, Jiang-Kai Qiu, Guigen Li and Kai Guo. After we received your comments, we have made following efforts on revising this paper. The appropriate changes have been made and marked with yellow highlight both in the revised manuscript and supporting info.

Reviewers' comments:

Reviewer #1 (Remarks to the Author):

Comments:

Guo and coworkers report a strategy for merging photoredox-mediated defluorofunctionalization of trifluoromethylarenes with heteroaryl migration to produce difluorobenzyl ketone products. The reaction employs conditions similar to König's work on the addition of trifluoromethylarenes to *N*-aryl acrylamide acceptors, but instead uses alkene acceptors that transfer *N*-heteroaryl groups to the radical adduct. The reaction proceeds under mild conditions but requires 2 equivalents of the alkene acceptor. Mechanistically, this is a nice extension of previous work in photoredox-enabled trifluoromethylarene defluoroalkylation chemistry and should expand the scope of accessible difluorobenzyl products. The authors show a range of heteroaryl groups that transfer (primarily 1,3-azole derivatives) and also demonstrate a few types of trifluoromethylarenes that this reaction works for. Mechanistic studies are included that support the proposed SET and radical-based mechanism.

Overall, this work is a welcomed extension of trifluoromethylarene functionalization chemistry but the extremely limited scope of ArCF₃ substrates reduces the utility of this method in its current state. Only 4-cyano and 4-sulfonamide benzotrifluorides participate in this reaction; this is a much narrower scope than reported by König or Jui, and it is unclear why this reaction does not work for other

substrate classes. The authors should either expand the scope of ArCF₃ partners or show that their products can be further derivatized. In the abstract, they state that the “products could be further transformed into other valuable compounds” but do not show this at all. I believe one of these two improvements should be made before publishing this work.

Response: Thanks for your kindly reminder. According to this reviewer’s valuable suggestion, we tried our best to further expand the scope of this transformation. Firstly, the para-cyano-substituted benzotrifluorides (**4o**), the ortho- and meta-cyano-substituted benzotrifluorides (**4p**, **4q**), and aldehyde substituted benzotrifluorides (**4r**) reported by König and Jui were investigated for this transformation, and the results indicated that the desired products could not be obtained smoothly. We speculated that the limited range of the ArCF₃ substrate is due to the electron-withdrawing ability of the substrate. Some substrates have insufficient electron-withdrawing ability and the reaction cannot proceed smoothly. At the same time, we also have tried to expand the scope of ArCF₃ partners. To our delight, the substrate **2g** was compatible well in this transformation and the corresponding product **4g** could be yielded in moderate yield. However, other substrates such as **2s**, **2t**, **2u** **2v**, **2w**, **2x** were proved not to be compatible with the reaction, and the yields of desired products (**4s**, **4t**, **4t**, **4v**, **4w**, **4x**) were trace basically (2%~ 3%, detected by ¹⁹F NMR). Although we have done our best but failed to obtain the perfect NMR data of desired products.

The specific data is as follows :

Expand the scope of ArCF₃ partners

Moreover, we also tried to show that our products could be further derivatized. To our delighted, we conducted the derivative reaction successfully. Under simple redox conditions, the product **3a** was converted to the corresponding aldehyde, and the desired product compound **9** could be obtained in the isolated yield of 32% over 3 steps. The corresponding revisions have been made both in the manuscript and supporting info.

Preparation of compound **9**:

General preparation procedure: compound **3a** (0.2 mmol), activated 4 Å molecular sieves powders (300 mg), and anhydrous DCM (2 mL) was stirred at rt for 10 min, then Me₃OBF₄ (0.5 mmol) was added. After stirred at rt for 2 h, another batch of Me₃OBF₄ (0.5 mmol) was added to the suspension, which was continued to react until **3a** was consumed as determined by TLC. The reaction was concentrated without filtering off the molecular sieves to give the crude *N*-methylbenzothiazolium salt. The residue was redissolve in MeOH (2 mL), which was then cooled to 0°C and added

NaBH₄ (0.3 mmol). Another batch of NaBH₄ (0.2 mmol) was added to the reaction until the starting material had been consumed as determined by TLC. The mixture was diluted with acetone, filtered through a pad of Celite, and concentrated to give the crude benzothiazolines. To a vigorously stirred solution of the crude benzothiazolines in CH₂Cl₂ (0.6 mL) and CH₃CN (3 mL) were added H₂O (0.36 mL) followed by AgNO₃ (0.6 mmol). The mixture was stirred at rt (monitored by TLC), and then diluted with 1 M phosphate buffer at pH 7 (0.2 mL). After stirred for 15 min, the reaction mixture was diluted with 1 M phosphate buffer at pH 7 (5 mL) and partially concentrated to remove CH₃CN. The suspension was extracted with EtOAc, and the combined organic layers were dried over Na₂SO₄, filtered through a pad of Celite, and concentrated. The residue was purified by flash column chromatography on silica gel to afford compound **9** (32% over 3 steps).

Compound **9**, 32% (Isolated yield). ¹H NMR (400 MHz, Chloroform-*d*) δ 9.68 (s, 1H), 7.92 (d, *J* = 8.1 Hz, 2H), 7.74 (d, *J* = 8.1 Hz, 2H), 7.64–7.55 (m, 3H), 7.48 (d, *J* = 7.7 Hz, 2H), 3.12–2.95 (m, 2H), 2.80–2.74 (m, 1H), 2.23–2.15 (m, 2H), 2.05–1.98 (m, 2H). ¹³C NMR (100 MHz, Chloroform-*d*) δ 201.5, 133.4, 132.6, 128.7, 128.0, 35.0, 29.3, 27.2, 23.4. ¹⁹F NMR (376 MHz, Chloroform-*d*) 94.6, 95.2, 95.7, 96.4. HRMS [ESI] calcd for C₂₀H₁₇F₂NO₂ [M+Cl]⁻ 376.0921, found 376.0947.

¹H NMR (400 MHz, CDCl₃) of **4g**

¹⁹F NMR (376 MHz, CDCl₃) of **4g**

^{13}C NMR (100 MHz, CDCl_3) of **4g**

HRMS of **4g** [ESI] calcd for $\text{C}_{26}\text{H}_{19}\text{ClF}_2\text{N}_2\text{OS}$ $[\text{M}+\text{Na}]^+$ 503.0767, found 503.0671.

$^1\text{H NMR}$ (400 MHz, CDCl_3) of **4t**

$^{19}\text{F NMR}$ (376 MHz, CDCl_3) of **4t**

The HRMS of **4t** [ESI] calcd for C₂₇H₂₂F₂N₂OS [M+Na]⁺483.1313, found 483.1225.

¹H NMR (400 MHz, CDCl₃) of **4v**

^{19}F NMR (376 MHz, CDCl_3) of **4v**

The HRMS of **4v** [ESI] calcd for $\text{C}_{31}\text{H}_{34}\text{F}_2\text{N}_2\text{O}_3\text{S}_2\text{Na}$ $[\text{M}+\text{Na}]^+$ 607.1871, found 607.1762.

$^1\text{H NMR}$ (400 MHz, CDCl_3) of **4w**

$^{19}\text{F NMR}$ (376 MHz, CDCl_3) of **4w**

The HRMS of **4w** [ESI] calcd for $C_{33}H_{30}F_2N_2O_3S_2Na$ $[M+Na]^+$ 627.1558, found 627.1434.

1H NMR (400 MHz, $CDCl_3$) of **4x**

^{19}F NMR (376 MHz, CDCl_3) of **4x**

The HRMS of **4x** [ESI] calcd for $\text{C}_{31}\text{H}_{32}\text{F}_2\text{N}_2\text{O}_3\text{S}_2\text{Na}$ $[\text{M}+\text{Na}]^+$ 605.1715, found 605.1593.

$^1\text{H NMR}$ (400 MHz, CDCl_3) of **9**

$^{19}\text{F NMR}$ (376 MHz, CDCl_3) of compound **9**

^{13}C NMR (376 MHz, CDCl_3) of compound **9**

The HRMS of **9** [ESI] calcd for $\text{C}_{20}\text{H}_{17}\text{F}_2\text{NO}_2$ [$\text{M}+\text{Cl}$]-376.0921, found 376.0947.

Question: For the radical trapping experiments shown in Figure 4, it would be very helpful to include what products were observed and in what quantity. Only stating that the coupling product was not obtained is of limited value, and including approximate yields of trapped intermediates would also be useful (compared to just stating that the adduct was “detected by HRMS”). Similarly, is F–Bpin observed as a byproduct of

this reaction?

Response: Thanks for your kindly reminder. According to this reviewer's valuable suggestion, we analyzed the reaction solution by ^{19}F NMR and HRMS and failed to find the byproduct F-Bpin (Exact Mass: 146.0914, $+\text{H}^+147.0987$). We also study the reaction between TMP and B_2Pin_2 , the signal at 24.3 ppm in the ^{11}B NMR spectrum represents the neutral amino-borane species and appears whenever TMP and B_2pin_2 are present. However, some of the previous reported works on the proposal mechanism such as *J. Am. Chem. Soc.* **141**, 13203–13211 (2019), *J. Am. Chem. Soc.* **139**, 18444–18447 (2017) could support our views. However, when we tried to expand our scope of ArCF_3 partners, we can not to get the desired while the F-byproduct was observed. So, we speculate that the reaction may produce F-by-products, the control mechanism is not yet clear, we will continue to study this research.

The specific modifications are as follows:

^{11}B NMR of the reaction between TMP (0.3 mmol) and B_2Pin_2 (0.3 mmol) in CD_2Cl_2 (1.0 mL) for 24 h under Ar atmosphere.

^{19}F NMR of the reaction solution

HRMS of the reaction solution

¹H NMR (400 MHz, CDCl₃) of **10**

The HRMS of **10** [ESI] calcd for C₂₀H₁₉CINOSNa [M+Na]⁺ 392.0637, found 392.0651.

¹H NMR (400 MHz, CDCl₃) of **11**

¹³C NMR (100 MHz, CDCl₃) of **11**

The HRMS of **11** [ESI] calcd for $C_{27}H_{20}F_4N_2OS$ $[M+H]^+$ 497.1305, found 497.1211.

Question: The authors state on line 205 that the reaction features “unique regioselectivity/stereoselectivity”. What does this mean? I don’t think this reaction has any stereochemical features to it at all and “unique regioselectivity” is meaningless if it is not referenced to a specific feature/other methods.

Response: Thanks for your kindly reminder. According to this reviewer’s valuable suggestion, we have re-checked line 205 and the inaccurate sentence “unique regioselectivity/stereoselectivity” have deleted. We are sorry for this careless mistake and the corresponding revise have been made in the revised manuscript.

Question: In the Supporting Information, it would be helpful to include the splitting patterns observed in the F NMR spectra and to include the IR data of the products.

Response: Thanks for your kindly reminder. According to this reviewer’s valuable suggestion, we have checked the IR data of the products and splitting patterns observed in the F NMR spectra, and the corresponding data have been made in Supporting Information (details see the revised Supporting Information)

Reviewer #2 (Remarks to the Author):

In this manuscript, Guo and co-workers have disclosed a photocatalytic radical defluoroalkylation of electron-deficient trifluoromethylarenes of unactivated alkenes via distal heteroaryl ipso-migration, which is originally developed by Zhu (JACS, 2017, 139, 1388). Surprisingly, this seminal paper was not cited herein. Frankly speaking, this work is a smart combination of two interesting concepts, C(sp³)-F activation and remote functional group migration. As a result, the reaction achieves a sequence of C-F and C-C bonds scission under mild conditions, which are generally challenging issues in synthetic chemistry. The substrate scope is reasonably broad, and some experiments have been performed to elucidate the possible mechanism. Because the overall transformation merges two current hot topics, it would attract broad interest from synthetic chemists. In sum, this manuscript could be considered for publication in Commun. Chem. but after major revisions.

Response: Thanks for this reviewer's high comments. It is our honor to get such a good comment, we will try our best to do better in the future. Meanwhile, we are so sorry to make such a careless mistake, and the originally report developed by Zhu's group (JACS, 2017, 139, 1388) have been added in the revised manuscript (details see the revised manuscript).

The specific modifications are as follows:

19. Z. Wu & D. Wang. Chemo- and Regioselective Distal Heteroaryl *ipso*-Migration: A general protocol for heteroarylation of unactivated alkenes. *J. Am. Chem. Soc.* **139**, 1388-1391 (2017).

Question: 1) Page 2, Line 44, ref. 14. Gschwind and König's work should be corrected with JACS 2017, 139, 18444. Page 2, Line 46, ref. 15. The related literature on photoredox-catalyzed monoselective defluorinative functionalization should also be corrected: JACS 2019, 141, 13203. Page 4, Line 57-61, the seminal literatures or reviews on distal heteroaryl migration should be cited, e.g. JACS 2017, 139, 1388; ACIE 2018, 57, 1640; ACIE 2018, 57, 17156; Org. Chem. Front. 2018, 5, 1896; Tetrahedron Lett. 2018, 59, 1328; CJC, 2019, 37, 171, etc.

Response: Thanks for your kindly reminder, we are so sorry for making these mistakes. According to this reviewer's valuable suggestion, we have modified and corrected the format of the reference section. Ref. 14. Gschwind and König's work has been corrected with JACS, 2017, 139, 18444. The related literature on photoredox-catalyzed monoselective defluorinative functionalization has also been corrected with JACS 2019, 141, 13203. Page 4, Line 57-61, the seminal literatures or reviews on distal heteroaryl migration have been cited, e.g. JACS 2017, 139, 1388; ACIE 2018, 57, 1640; ACIE 2018, 57, 17156; Org. Chem. Front. 2018, 5, 1896; Tetrahedron Lett. 2018, 59, 1328; CJC, 2019, 37, 171, etc.

The specific modifications are as follows:

14. Chen, K., Berg, N., Gschwind, R. & König, B. Selective single C(sp³)-F bond cleavage in trifluoromethylarenes: merging visible-light catalysis with Lewis acid activation. *J. Am. Chem. Soc.* **139**, 18444–18447 (2017).
15. Vogt, D. B., Seath, C. P., Wang, H. & Jui, N. T. Selective C-F functionalization of unactivated trifluoromethylarenes. *J. Am. Chem. Soc.* **141**, 13203–13211 (2019).
19. Wu, Z. & Wang, D. Chemo- and Regioselective Distal Heteroaryl *ipso*-Migration: A General Protocol for Heteroarylation of Unactivated Alkenes. *J. Am. Chem. Soc.* **139**, 1388-1391 (2017).
20. Wu, X., Wang, M., Huan, L., Wang, D., Wang, J & Zhu, C. Tertiary-Alcohol-Directed Functionalization of Remote C(sp³)-H Bonds by Sequential Hydrogen Atom and Heteroaryl Migrations. *Angew. Chem. Int. Ed.* **57**, 1640-1644 (2018).
21. Wang, M. & Wu, Z. Azidoheteroarylation of unactivated olefins through distal heteroaryl migration. *Org. Chem. Front.* **5**, 1896-1899 (2018).

22. Yu, J., Wu, Z. & Zhu, C. Efficient Docking–Migration Strategy for Selective Radical Difluoromethylation of Alkenes. *Angew. Chem. Int. Ed.* **57**, 17156-17160 (2018).
23. Wu, X., Wu, S. & Zhu, C. Radical-mediated difunctionalization of unactivated alkenes through distal migration of functional groups. *Tetrahedron Lett.* **59**, 1328-1336 (2018).
24. Wu, X & Zhu, C. Recent Advances in Radical-Mediated C—C Bond Fragmentation of Non-Strained Molecules. *Chinese Journal of Chemistry* **37**, 171-182 (2019).

Question: 2. There are many grammatical errors and typos. The manuscript should be thoroughly polished. For example:

Page 7, Line 116, “thiophene (1k) or naphthalene (1l)” should be changed to “thienyl (1k) or naphthyl (1l)”, and related description should also be corrected in Fig. 2. Page 8, line 122, the compound is “benzoxazole (1o)” not “benzimidazole (1o)”. Other format errors in Page 17, Line 258 and Line 265; Page 18, Line 281 should be corrected.

Response: Thanks for your kindly reminder, we are so sorry for making these mistakes. We invited the native English speaker to polish the language of the manuscript. More details could be found in the revised manuscript. The other questions mentioned were also revised. In Page 7, Line 116, “thiophene (1k) or naphthalene (1l)” has been replaced by “thienyl (1k) or naphthyl (1l)”, and related description has also been corrected in Fig. 2. Other format errors in Page 17, Line 258 and Line 265; Page 18, Line 281 have been corrected. More details could be found in the revised manuscript. In Page 8, line 122, “benzimidazole (1o)” has been corrected by “benzoxazole (1o)”.

The specific modifications are as follows:

In addition to phenyl derivatives, other aryl groups such as **thienyl (1k)** and **naphthyl (1l) groups** were suitable for this intramolecular heteroaryl migration, although lower yields were obtained (**3k** and **3l**).

The *N*-containing heteroaryl groups could also be extended to **benzoxazole (1o)**, pyridine (**1p**), and **thiazole (1q and 1r)**, although with slightly decreased yields (**3o–3r**).

11. Stahl, T., Klare, H. F. T. & Oestreich, M. C(sp³)-F bond activation of CF₃-substituted anilines with catalytically generated silicon cations: spectroscopic evidence for a hydride-bridged Ru-S dimer in the catalytic cycle. *J. Am. Chem. Soc.* **135**, 1248-1251 (2013).

13. Luo, C. & Bandar, J. S. Selective Defluoroallylation of trifluoromethylarenes. *J. Am. Chem. Soc.* **141**, 36, 14120-14125 (2019).

27. Zheng, M., Yuan, X., Cui, Y.-S., Qiu, J.-K., Li, G.-g. & Guo, K. Electrochemical sulfonylation/heteroarylation of alkenes via distal heteroaryl *ipso*-migration. *Org. Lett.* **20**, 7784-7789 (2018).

Question: 3. Page 7, Line 120, no electron-withdrawing substituents have been tested, so the conclusion “the substituted benzothiazoles were migrated smoothly in spite of the electronic characters” is not accurate.

Response: Thanks for your kindly reminder, we are so sorry for making this mistake. According to this reviewer’s valuable suggestion, the inaccurate sentence “the substituted benzothiazoles were migrated smoothly in spite of the electronic characters” in Page 7, Line 120 have been deleted.

4. Fig 5, the authors should explain the step from G to J. Which species acts as the single-electron oxidant?

Response: This is a good question. We have considered contributions to the mechanism from a radical chain. Based on the previous literatures and the Fluorescence Quenching Experiment, we propose a tentative mechanism for the reaction which begins with visible light excitation of **Ir(III)*** or **A** act as the single-electron oxidant. Intramolecular interception of the alkyl radical by the C–N double bond of heteroaryl leads to a spiro-bicyclic *N*-centered radical intermediate **J**. Some reported works about the proposal mechanism {*J. Am. Chem. Soc.* **141**, 13203–13211 (2019); *J. Am. Chem. Soc.* **139**, 18444–18447 (2017); *J. Am. Chem. Soc.* **139**, 1388-1391 (2017); *Angew. Chem. Int. Ed.* **57**, 1640-1644 (2018)} could support our views. The corresponding revisions have been made in the revised manuscript.

Reviewer #3 (Remarks to the Author):

In the manuscript, the authors reported a tandem C(sp³)-F and C-C bonds functionalization by visible-light photoredox catalysis and Lewis acid activation. The strategy provides a useful method for construction of multiple functionalized α , α -difluorobenzyl ketone in moderate to good yields under mild conditions. The authors also proposed a reasonable mechanism for this transformation. This is a meaningful research. However, The English writing should be polished, and some spelling and grammatical mistakes need to be corrected. For example, ‘theombination’ in Line 202 might be ‘the combination’. In short, this manuscript is required an appropriate revision before to be published.

Response: Thanks for this reviewer’s high comments. We invited the native English

speaker to polish the language of the manuscript. 'theombination' in Line 202 has been replaced by 'the combination'. Furthermore, we have re-checked the whole text carefully and some grammatical mistakes have been corrected. More details could be found in the revised manuscript.

If you have any more questions regarding the revision, please feel free to contact me.

Best regards

Dr Kai Guo

Professor, State Key Laboratory of Materials-Oriented Chemical Engineering

Dean, College of Biotechnology and Pharmaceutical Engineering

Nanjing Tech University

No. 30 Puzhu South Road, Nanjing, Jiangsu Province, China

Tel: 0086 25 58139901

0086 13851647545

Email: kaiguo@njtech.edu.cn

REVIEWERS' COMMENTS:

Reviewer #1 (Remarks to the Author):

The authors have addressed the reviewer comments satisfactorily and I support publication. Although the substrate scope for the trifluoromethylarene coupling partner remains limited, the overall reaction is still an interesting example of monoselective C–F functionalization. I have just a couple small questions for the authors to clarify in their revised manuscript from Figure 4.

In Figure 4a, are any TEMPO-trapped products observed? The yield should be stated in the manuscript, if observed. What is the yield of 8 in Figure 4c?

Reviewer #2 (Remarks to the Author):

In this revision, the authors addressed all my previous concerns. Thus, I support it to be accepted for publication in current form.

Dear reviewers:

Thank you very much for your valuable suggestions on our manuscript entitled "Photocatalytic radical defluoroalkylation of unactivated alkenes *via* distal heteroaryl *ipso*-migration" (COMMSCHEM-19-0372A) by Xin Yuan, Kai-Qiang Zhuang, Yu-Sheng Cui, Long-Zhou Qin, Qi Sun, Xiu Duan, Lin Chen, Ning Zhu, Guigen Li, Jiang-Kai Qiu, and Kai Guo. After we received your comments, we have made following efforts on revising this paper. The appropriate changes have been made used the 'Track changes' feature to make the process of accepting our manuscript more efficient in the word.

REVIEWERS' COMMENTS:

Reviewer #1 (Remarks to the Author):

The authors have addressed the reviewer comments satisfactorily and I support publication. Although the substrate scope for the trifluoromethylarene coupling partner remains limited, the overall reaction is still an interesting example of monoselective C–F functionalization. I have just a couple small questions for the authors to clarify in their revised manuscript from Figure 4.

In Figure 4a, are any TEMPO-trapped products observed? The yield should be stated in the manuscript, if observed. What is the yield of **8** in Figure 4c?

Response: Thanks for your kindly reminder. Actually, we could not observe the TEMPO-trapped products **5** when we conducted the reaction with the addition of TEMPO. Instead, the unexpected product **5a** can be isolated with the yield of 11% (Fig. 4a). We speculated that TEMPO could also react with substrate **1a** as a free radical donor. However, the TEMPO-trapped products **5** could also be detected by the HRMS when we conducted the reaction without the addition of substrate **1a** (for more details, see the Supplementary Figure 4.).

The specific steps are as follows:

4-(Trifluoromethyl)benzotrile **2a** (0.1 mmol, 17.1 mg, 1.0 eq.), TEMPO (2.0 eq., 31.2 mg), B₂pin₂ (0.3 mmol, 76.2 mg, 3.0 eq.) and *fac*-Ir(ppy)₃ (0.001 mmol, 0.7 mg, 1.0 mol%) were added into a 10 mL snap vial equipped with a stirring bar. The vial was purged with Ar for three times via syringe needle. Then TMP (0.3 mmol, 42.3

mg, 51 μ L, 3.0 eq.) and dry THF (1.0 mL) were added sequentially by syringes. The reaction mixture was irradiated through the bottom side of the vial by Blue LEDs at 25°C. Then the reactions were concentrated under reduced pressure. The crude residues were analyzed by HRMS. The TEMPO trapped aryldifluoromethyl radical **5** could be detected by HRMS. HRMS (ESI) m/z $[M+Na]^+$: 331.1586.

Fig. 4a Trapping experiment by TEMPO

b) Radical Trapping Experiment by Tempo without 1a

Supplementary Figure 3. HRMS (ESI) data of **5a**

Supplementary Figure 4. HRMS (ESI) data of 5

In Figure 4c, the crude residues were analyzed by ¹⁹F NMR (42.1 mg of PhOCF₃ as internal standard) and HRMS (ESI). The conversion of **2a** was 38%. A mixture of saturated and unsaturated trapping products **8** and **8'** were obtained only in 8% yield. The appropriate changes have been made in the revised manuscript. ¹⁹F NMR (376 MHz, CDCl₃) δ 81.6 (unsaturated), 87.7 (saturated). HRMS (ESI) m/z [M+Na]⁺: 354.9486 (unsaturated), [M+H]⁺: 334.1413 (saturated).

4c) Radical Trapping Experiment by 1,1-Diphenylethylene

Supplementary Figure 6. HRMS (ESI) data of 8' (unsaturated)

Supplementary Figure 7. HRMS (ESI) data of 8 (saturated)

Supplementary Figure 105. Radical Trapping by 1,1-Diphenylethylene, ^{19}F NMR of cruder mixture (376 MHz, CDCl_3)

Reviewer #2 (Remarks to the Author):

In this revision, the authors addressed all my previous concerns. Thus, I support it to be accepted for publication in current form.

Response: Thank you for your highly comments. It is our honor to get such a good comment, we will try our best to do better in the future.

Best regards

Dr Kai Guo

Professor, State Key Laboratory of Materials-Oriented Chemical Engineering

Dean, College of Biotechnology and Pharmaceutical Engineering

Nanjing Tech University

No. 30 Puzhu South Road, Nanjing, Jiangsu Province, China

Tel: 0086 25 58139901

0086 13851647545

Email: kaiguo@njtech.edu.cn